# Training Conditions and Psychological Health: Eating Behavior Disorders in Spanish High-Performance Women’s Olympic Wrestling Athletes—A Qualitative Study

**DOI:** 10.3390/ijerph20032441

**Published:** 2023-01-30

**Authors:** Marina Rueda Flores, Daniel Mon-López, Javier Gil-Ares, Javier Coterón

**Affiliations:** 1Facultad de Ciencias de la Actividad Física y del Deporte (INEF—Sports Department), Universidad Politecnica de Madrid, 28040 Madrid, Spain; 2Facultad de Ciencias de la Actividad Física y del Deporte (INEF—Social Sciences Applied to Physical Activity, Sport and Leisure Department), Universidad Politecnica de Madrid, 28040 Madrid, Spain

**Keywords:** psychological aspects, Olympic wrestling, female wrestlers, eating disorders, qualitative research

## Abstract

(1) Background: the aim of this study was to determine the factors that affect the occurrence of behaviors related to possible eating disorders in Spanish high-performance Olympic wrestling athletes. (2) Methods: The sample consisted of 22 elite female wrestlers selected through purposive sampling with inclusion criteria of (i) having been a national champion, (ii) having been part of the Spanish team, and (iii) suffering or having suffered from an eating disorder. The semistructured interviews were conducted online and lasted between 20 and 40 min. A statistical analysis was performed with NVivo10 software. (3) Results: the athletes showed a series of issues grouped into three main themes, which were divided into the following categories—firstly, the reasons why wrestlers lose weight; secondly, the inadequate procedures they use; and, finally, the reference persons involved in the process. (4) Conclusions: The training conditions in high-performance sports have psychological and behavioral effects on wrestlers. Wrestlers have to move down to lower categories; however, they do not take into account how this practice influences their health when using inadequate procedures. Rapid and significant weight loss produces negative effects, especially in the female population, generating an incidence of eating disorders. The information obtained provided elements of interest for reflection on possible solutions to prevent existing eating disorders.

## 1. Introduction

Eating disorders (EDs) are severe disturbances in eating, weight control behaviors, and body image. In addition to so-called unspecified EDs, the American Psychiatric Association [1] distinguishes three types of clinical eating disorders: anorexia nervosa, bulimia nervosa, and binge eating disorder. EDs have been on the rise in recent years due to many factors, including changes in our society in relation to health and sports [2].

According to the American Psychiatric Association (2014) [3], EDs are determined by a persistent disturbance in eating or eating-related behaviors that leads to a significant impairment in physical health or psychosocial actions. Although these health problems can affect the entire population, athletes are at an increased risk of developing EDs and inadequate exercise behaviors [4]. In this line, Moriarty and Moriarty (1993) [5] and Martinez (2012) [6] considered the psychological characteristics of the athlete and the pressures from society, one’s coach, one’s teammates, and the sport itself as factors that can facilitate the appearance of EDs. Additionally, these EDs can have adverse consequences on athletic performance, well-being, health, and athlete identity [7,8].

Regarding sports, weight category sports are those in which weight defines the participant’s category. In this line, the previous literature has shown that weight category sports (e.g., judo, wrestling, rowing, and boxing), sports related to body aesthetics (e.g., gymnastics, skating, and synchronized swimming), and endurance sports (e.g., long distance athletics) seem to be susceptible to an increased risk of developing EDs [9]. However, other studies have reported differences between the athletes of weight category sports and the athletes of other sports in their scores on validated questionnaires related to EDs, such as the Eating Attitudes Test or the Eating Disorder Inventory [10].

Inside weight category sports, we can find the Olympic wrestling sport. Olympic wrestling is a combat sport in which each wrestler attempts to defeat his/her opponent without the use of strikes. Olympic wrestling is one of the oldest sports on record, and, throughout history, it has included some rules to allow sports standardization. One of these rules is the division of wrestlers into age and weight categories to create a level playing field [11]. In consequence, it is often observed that wrestlers aim to be in the lowest weight category possible. Thus, athletes usually think that they will have a relative advantage when facing lighter and less strong competitors [8]. In addition, Engels et al. (2003) [12] found that wrestlers scored high on weight loss tendency, dietary restraint, and purging behaviors compared to other athletes in other sports.

In order to lose weight, some wrestlers attempt to lose weight rapidly, most often through drastic weight drops or pathological weight loss practices with subsequent uncontrolled intake, and they complete their weight loss plan by using some form of purging (vomiting, the use of laxatives or diuretics, or even excessive exercise) or by using dehydration methods (saunas, sauna suits, or salt baths) [13]. Moreover, there seems to be a relationship in wrestling and combat sports between inadequate weight loss practices and physical, athletic, and cognitive performance problems [14,15]. Thus, the studies in this area have even pointed to a high risk of having EDs reaching a prevalence of 26% [16].

Regarding gender, studies state that the number of EDs has increased in recent years, especially in women [9]. In general, it seems that the prevalence is higher in women than in men [17]. This fact could be related to factors such as low self-esteem, body image, social networks, or the pressure received both by themselves and by their own environment [8]. Accordingly, the behaviors of athletes, such as the manipulation of food and the use of inadequate weight control methods, may contribute to the development of EDs and serious health problems. The previous literature revealed that 46.15% of female wrestlers presented some type of ED, such as anorexia or bulimia, suggesting the need for more training and intervention programs to improve the health of these athletes [18].

Although EDs can have serious consequences on athletes’ present and future, there seems to be scarce literature that explains in depth the causes of their appearance and the process in which EDs are developed in athletes, especially in high-performance female Olympic wrestlers. In consequence, the main objective of this study was to determine the factors that affect the appearance of behaviors related to possible EDs in Spanish high-performance Olympic wrestling athletes.

## 2. Materials and Methods

### 2.1. Study Design

An important methodological issue in the study of EDs is the fact that individuals may be dishonest about themselves because they experience feelings of shame, self-blame, or fear and, therefore, attempt to hide their inappropriate eating and dieting patterns [19]. In fact, evidence of significant underreporting of EDs among athletes has been found when comparing questionnaire data with clinical interviews. In this regard, various studies have pointed to the appropriateness of qualitative methodology. Studies state that a descriptive phenomenology approach focuses on describing the participant’s lived experience to better understand a phenomenon [20]. Research design from a phenomenological paradigm can help to better understand the beliefs and motivations that govern subjects’ behaviors from their personal experience and perspective. It also seems essential for researchers to develop trusting relationships with subjects in order to obtain quality information [21].

### 2.2. Participants

National coach of the Spanish women’s Olympic wrestling team was contacted, and she provided us with the contact details of the 30 elite Spanish female wrestlers who were members of the official national team at that moment. Of the total number of athletes, 2 of them declined the invitation, and 6 of them were discarded according to the inclusion criteria or because they had never suffered an ED or had never had an unhealthy relationship with food, leaving a final sample of 22 female wrestlers. After conducting the 22 interviews, there was no need to conduct more because the saturation level was amply reached. The mean age at the time of data collection was 20.82 ± 2.79 years old. The rest of the participants’ descriptive characteristics are shown in Table 1.

Three inclusion criteria were used to identify subjects who could provide relevant information to answer the research questions due to having a broad perspective of the situation [19]: (1) being elite, or having been champion of Spain in any age category, and (2) having been part of the Spanish national team, participating in at least one international championship, and suffering or having suffered from an ED.

### 2.3. Material and Data Collection

A semistructured interview (Appendix A) was developed by two of the authors, the first being a high-level professional female wrestler with many years of experience in the field and with a background in qualitative research and the second being a sports scientist specialized in qualitative methodology and a professor of this subject with many years of experience in qualitative research, and it was helped by using the triangulation strategy with an expert external to the study in order to be able to gain in-depth insight into the personal experience of the female wrestlers [22]. It was developed from themes and questions based on previous research related to EDs in high-performance female athletes. The script was structured around three blocks. First, introduction and general warm-up questions. In the second block, the interviewee was asked to talk about her personal history: her personal experience in terms of her ED, Olympic wrestling, other influential factors, help received, possible solutions, etc. Finally, in the third block, they were asked to indicate what measures they thought should be taken to avoid future cases and what would be the appropriate steps to solve the cases that already exist.

Prior to the start of the study, two pilot interviews were conducted with volunteer wrestlers to adjust the questions and the script. These were conducted jointly by the lead author and another author, a sports scientist specializing in qualitative methodology and a professor of this subject with many years of experience in qualitative research, for supervision. All the interviews in the study were conducted by the lead author, who had experience as an interviewer in previous studies, in order to guarantee the stability of the data during the research [23].

All interviews were conducted by the lead author, who had experience as an interviewer in previous studies. The pilot interview was conducted jointly with another author, a sports scientist specializing in qualitative methodology and a professor of this subject with many years of experience in qualitative research, for supervision.

All participants were given a signed consent form, and it was explained to them that participation was strictly confidential and voluntary. The interviews were conducted online by video call due to the special conditions caused by the COVID-19 pandemic and lasted between 20 and 40 min, and we used audio recording.

Before starting the interview, the main objective of the research was explained to the wrestlers so that they knew the type of research in which they were going to participate. The voluntary and confidential nature of the study was repeated, and it was explained to them that they could withdraw from the study at any time without any consequences. They were asked for permission to record the interview, and it was made clear to them that any details that could lead to the disclosure of their identity would be excluded from the results. In addition, explicit mention was made of the possibility of consulting the first author or having access to a psychologist from the Spanish Federation after participation in case the participants had any questions or needed professional support. All the aforementioned information was recorded on the informed consent form that was signed by all the participants.

The interview was conducted following a phenomenological approach [24]. Phenomenological research uses qualitative and naturalistic approaches to inductively and holistically understand human experience in specific contexts [25]. In this study, this concerned the development of athletes’ EDs in the contexts of everyday life and sports. Whenever participants appeared to hold back or give noncommittal responses, probing and detail-oriented questions were asked to help identify influences on their eating. Field notes were taken right after the interview.

The protocol for this study was approved by the Ethics Committee of the Universidad Politécnica de Madrid. To ensure anonymity, specific identifying details of participants and clubs were withheld.

### 2.4. Data Analysis

Each of the audio-recorded interviews was transcribed by the principal investigator with the collaboration of two qualitative researchers external to the study onto a Word^®^ document. Thematic analysis method [26] was conducted to analyze the data. First, a superficial reading of the documents was carried out in order to become familiar with the data and to determine the recording units. Then, labelling was carried out with an open and inductive coding system, searching for the main themes emerging from the participants’ discourse and locating all the recording units obtained. The total number of coded recording units was 1140. Themes initially set out were discussed, modified, and reorganized until stability was checked and a consensus was reached on all themes and dimensions. This process of triangulation, or convergent validation, was carried out in various meetings of the research team in order to guarantee the validity of the results and reduce possible problems of bias [27]. The entire coding and categorization process was carried out using QSR NVIVO v1.5.1 (940) software.

## 3. Results

The most relevant elements resulting from the discourse analysis of the interviews conducted are presented below. Three main themes emerged: (1) the conditioning factors of inadequate weight loss, (2) the inadequate procedures that wrestlers carry out for such weight loss, and (3) the role of the reference people. In each of the themes, subthemes of interest were found for a better understanding of the wrestlers’ experiences (Table 2).

### 3.1. Reasons for Weight Reduction

#### 3.1.1. Limitation of Participation in International Championships

In major competitions, participation is limited to one representative from each country per weight category. This generates great competition among teammates in the same category. In some cases, the second-place finisher, if she is a high-level competitor, may change categories to qualify for the event in another category.


*“… my competition weight didn’t fit into the Olympic weights. So, of course, there was only one alternative. Well, two alternatives. Either go up one more weight category and try at that weight category or go to a lower weight and try that weight, we decided among all of us to go to a lower weight, which was 53 kg. I was competing at 59 kg…”*
(L09)

In national tournaments, there is no limitation of participants. The wrestlers are distributed in all the categories, resulting in greater representation and possibilities of awards. This means that some wrestlers must alter their weight to distribute competitors across the weights.


*“When there are more people on the team, we try to distribute the weights. If there are five weights and three of us are in similar weights, you still have to lose 2 kilos, the other one has to go up weighing a little less…”*
(L11)

#### 3.1.2. Body Changes: Development and Body Image

The process that leads a female wrestler to reach high performance usually begins before she reaches physical maturity. Weight and age are the factors that establish the categories in wrestling. Most athletes begin wrestling before adolescence in a certain weight category. Subsequently, a series of inevitable changes occur with normal physical maturation which may have consequences on one’s structure and body weight.


*“When I changed my age category, the minimum weight category was 50 kg, until then I wrestled at 43 kg and then I had to gain weight to wrestle with people who weighed the same. At that moment I realized that I had put on a lot of weight, I gained a lot of weight, and that’s when my problem began.”*
(L02)

In many cases, a struggle to remain within the same category begins despite the obvious physical changes resulting from normal physical development.


*“It was very hard, especially the first years, because you are in a growing age. Even if you eat a little bit, you gain a lot of weight, at least that’s what happened to me … And I had to stay at my weight, which was 46 kilos…”*
(L07)

Because wrestling is a sport that requires being in a weight class in order to compete, weight is associated with physique. For many female wrestlers, the number on the scale may reflect their physical condition in their view. They believe that if they are at their competition weight, they look good, but if not, they may believe that they look fat.


*“… if a person weighs 57 kilos and you see her looking very well, very slim, and that person gains two more kilos, she already sees herself and says: my God, I’ve gained a lot of weight, I’ve put on a lot of weight. It’s a continuous beating”*
(L07)

Regarding body image, one important aspect that we found was changes during puberty. During adolescence, the body develops and changes, which may generate a problem of self-acceptance for female wrestlers who start training at a young age.


*“… when I was 15 years old I started to have problems with my body image and I didn’t see myself correctly…”*
(L01)

Female wrestlers often have a strong sense of perfectionism that can result in high self-expectations. In addition to demanding high spot performance, female wrestlers may be critical in other ways, so they may strive to achieve perfection with their own physical image.


*“I am a very self-demanding person, so just as I am demanding with wrestling with my studies with everything, I consider that I was also very demanding with my image.”*
(L01)

We also found that, in the interviews, most of the female wrestlers tended to compare themselves physically with other female wrestlers or even with other sportswomen, which, in some cases, may have provoked discomfort with their body image.


*“It was constantly a physique comparison. Well I went up in weight class, I’m skinnier than this one or I’m stronger than this one. I want to be like this one. It was more focused on my image, but then it became a problem.”*
(L05)

### 3.2. Inadequate Procedures

The moment of the weigh-in influences the future of the season and the wrestlers’ possibilities of progressing and competing based on their weight. Consequently, the weigh-in can be stressful and can result in risky decisions regarding weight.


*“you live by and for, in my case, to making the weight.”*
(L05)

The stress generated by the need to reach a certain weight introduces a dynamic in athletes’ activities that affects different areas. This stress can generalize to other parts of athletes’ lives.

#### 3.2.1. Lack of Adequate Information

When it came to weight loss, most of the wrestlers reported that they did not seek or receive help.


*“… sometimes you’re at an age when you don’t want help. Or that at least the help they are giving you is not what you want.”*
(L06)

Due to not having adequate knowledge about nutrition, female wrestlers may attempt to manage weight using unsafe and unsustainable methods. For example, some wrestlers may believe that because the food weighs more, the result of eating such food will be greater weight gain.


*“I didn’t even know what quantities to eat, what to eat, or what to do to really lose weight.”*
(L13)

Sometimes wrestlers recognize that they do not know how to lose weight and entrust their weight loss to other people who are not specialists either (e.g., relatives), and they follow advice that is often well intended but highly problematic.


*“… I think that all of us have been told, okay, you have to lose weight, lose so many kilos and nobody has told us, okay, I’m going to make you a diet, or some habits so that you don’t really have to lose weight … no, they told you, well, the last day you jump in the sauna and that’s it.”*
(L11)


*“… there are a lot of people who give you advice and the advice creates problems for you.”*
(L08)

Within the wrestling environment, the younger and less experienced competitors look at how the more experienced wrestlers lose weight as a model for how to manage their weight. Problems may arise, however, when they look at more experienced wrestlers who are not managing their weight in healthy ways and then start imitating methods that are dangerous to their health, such as dehydration.


*“… at first you think, ouch, look at the big ones [female wrestlers]. You already see yourself a little bit reflected that you are in a high level competition team, when you see that you go out internationally, you see the team, you see the big ones[wrestlers] running and you already feel part of the team because you think you are doing something already more, as it were, important, by losing weight.”*
(L06)

#### 3.2.2. Risky Practices

Certain behaviors are introduced that are initially intended to ensure weight control but that later become habits that may pose a risk to athletes’ health. The risky practices that are carried out appear at two specific times: before the weigh-in, in order to reach the intended goal, and after the weigh-in, to compensate for the emotional stress and the physical stress on the body.


*“I spend a month in starvation, we go between training more and eating less and then you compete and you binge.”*
(L05)

Before the weigh-in, some of the female wrestlers interviewed confessed to having forced sudden weight loss using dangerous methods, such as extreme food restrictions at certain times, vomiting what was eaten to avoid weighing more, the use of laxatives and diuretics, etc.


*“There are behaviors that I thought weren´t there and they are. The use of diuretics, of laxatives, of things that I have not come to do, but that I thought nobody did.”*
(L05)


*“I used to lose 4 kilos in a week. Sweating, not drinking water, vomiting … however I could, but I would lose 4 kilos … I was very anxious or depressed. I started taking laxatives … I was terrible and they took me to the doctor.”*
(L06)

Restricting liquids is one of the most commonly used methods of weight loss due to its immediacy of results. Not only is drinking stopped but saunas, plastic suits, or salt baths are also used to facilitate water loss and, consequently, weight loss.


*“I hadn’t drunk water for a whole day and I was dead.”*
(L06)


*“… I exercised with lots and lots of clothes on … with plastic on.”*
(L08)

Given the ease and speed of the results, some of the female wrestlers reported to using these methods when they were warned that they were going to be weighed during the training period.

Adjusting menstruation with contraceptive methods so that it does not coincide with the competition period is another habit used by some female wrestlers to control their weight.


*“… if you take out your vaginal ring before you should, then your cycle starts before it should, so it is something that is not healthy, because you should have a regular cycle and you are changing it as it suits you.”*
(L11)

The ultimate goal is to reach the right weight by the precompetition weigh-in, but the dynamic does not end there. Many of the wrestlers reported binge eating right after an official weigh-in as compensation for the restrictions they had to apply previously.


*“… when you are in a championship or in the previous week, you can’t even with your body because you can practically only drink water and you eat very little, but after the weigh-in you feel terrible because you are bingeing on food and you start to look very overweight.”*
(L16)

Some of the wrestlers commented that they had even eaten until they vomited and then continued eating because they could not stop; they lost control. They knew that it was not good for competition, but they would still binge eat. A few talked about how badly they competed after doing this. Days after the weigh-in, some were still anxious about food and may even have gained significant weight because of the rebound effect, which they would then have to lose again, perpetuating the cycle.


*“… after the weigh-in is what I tell you, I ate a lot because of everything I had not been able to eat before and what I did eat, well, the night … that same night I felt super heavy. I was not able to really digest. And the next day I also woke up with a feeling, I don’t know how to describe it but of heaviness and obviously those were not the best circumstances to compete in.”*
(L11)

### 3.3. Role of Reference People

So far, we have talked about self-induced behaviors or behaviors performed by imitating the behaviors of other wrestlers. The role played by the people involved in the training process, such as coaches, psychologists, and nutritionists, is also of great importance.

#### 3.3.1. Sports Environment

Within the sports environment, female wrestlers usually trust their coaches, who, in many cases, give appropriate advice. However, in others, they follow what they have traditionally experienced in the wrestling world, and this can cause traditionally dangerous practices to be replicated because of the trust athletes have in the people who should set an example.


*“… as I had ignorance, because since you enter to compete, you lose weight how your partner tells you to, how the coach tells you to, you get dehydrated or whatever.”*
(L05)


*“… my coach has worked on that to compete, that is, to train already with a little dehydration because you compete better.”*
(L17)

In some cases, coaches may not be able to detect the problem that is brewing, downplaying behaviors that can lead to a disorder. It is usually because certain behaviors are commonplace and have become normalized within the wrestling world despite their potential risks.


*“I think that at the time, rather than helping, they made it worse. Because they were the ones who put the idea of weight in my head and made it seem like it was normal, so I preferred not to keep telling them about it.”*
(L12)


*“Nobody gave me tools to control anything, they didn’t even tell me, you have this problem. They [the coaches] wouldn’t even tell me you have an eating disorder. I didn’t know what was wrong with me. I knew it was wrong, but what was wrong with me? I didn’t know, I didn’t even ask myself because I thought it was normal…”*
(L6)

Many of the wrestlers pointed out that it was necessary to have the help and guidance of a specialized sports psychologist. The female wrestlers who reported having suffered with a disorder explained that if they had had this type of help, either they would not have fallen into this type of behavior, or they would have been able to be free of it sooner. However, despite help from clinical psychologists, many indicated that the problems persist.


*“… I have had a psychologist in this matter we have never gone into a deep conversation about it…”*
(L08)


*“… my family, my environment, my friends, but especially the psychologist. Psychological treatment seems like something fundamental to me.”*
(L20)

Regarding nutritional support, many of the wrestlers agreed that the help of a nutritionist who is an expert in weight loss for competitors is necessary. Many had usually gone to general nutritionists, having contradictory experiences between what was recommended for the general population and what was experienced in the high-energy-demand conditions of high-intensity competition training.


*“I approached the nutritionist to guide me or to control me and not to be told no fried food, no sweets, no battered food, it was something I already knew after 12 years of dieting … She was a nutritionist for a normal person with a sedentary lifestyle. I was expecting a lot more…”*
(L14)

#### 3.3.2. Social and Family Environment

The wrestlers reported often feeling misunderstood by the people around them who did not practice wrestling and were not part of this world, including family and friends. Many commented that, despite their best intentions, family, friends, and partners who offered support could actually be experienced negatively due to their lack of knowledge or specific training in competitive wrestling and weight management.


*“… my mother often times what she does to help me when I’m stressed because I can’t lose weight, is to plan what I have to eat to lose weight. And tell me that I’m going to make it, not to stress or worry.”*
(L10)


*“… my friends from the university, my lifelong friends, that I tell them and tell them and they say ho girl, but how are we going to lose 5 kilos? Wow! But, no, they have never been hungry…”*
(L05)

## 4. Discussion

The main objective of this study was to determine the factors that affect the appearance of behaviors related to possible eating disorders in Spanish high-performance Olympic wrestling athletes.

In order to create a better understanding, to facilitate appropriate prevention strategies, and to propose possible solutions, it is essential to address the problems with the current situation.

Following the same presentation structure as the results, we addressed the discussion to facilitate an understanding of the main findings.

### 4.1. Weight Control as a Determining Factor

The participants in this study strived for weight reduction, which conditioned, to a large extent, their behaviors. The desire to control weight has been identified in the current literature on the subject as one of the main risk factors for the development of eating disorders in elite athletes [28] and appears to be associated with certain psychological traits. For example, perfectionism has been studied as a cognitive style characteristic of people who develop EDs [29].

On the other hand, it was found that athletes who are more concerned about making mistakes in their sports practice and who have a high tendency to perfectionism develop more ED symptoms [30]. A relationship was also found between these behaviors and a lack of self-esteem [31].

Sometimes, the level of self-demand regarding their image leads them to fall into negative comparisons with other female wrestlers [9] and even with other athletes in other sports. This comparison is often another one of the causes that leads them to use different methods to improve their sports performance as well as their body image and shape [32]. The body dissatisfaction that leads to this constant comparison seems to be related to disordered eating behaviors [33].

The influence of the developmental age should also be considered. The inevitable physical development during this age implies a natural increase in weight. Sometimes, this fact is ignored, making great efforts to maintain a category that is no longer adequate to one’s new body constitution. In the specific literature on the origin of EDs, the appearance of a moment of change or crisis is pointed out as the main factor underlying the origin of the disorders [9]. The idea of staying at an inappropriate weight can sometimes be one’s own idea or one introduced by outsiders, whose opinions can exert undue influence on the athlete [6].

The competition structure and classification systems also emerge as determinants for weight control. The official weigh-in, before and after competition, generates unhealthy behaviors in the short term regarding food [34]. In the medium term, the limitation of participation in international championships forces wrestlers who do not obtain a place in a category to consider participating in another weight category, usually in a lower one, where they may have more possibilities [18,35]. This decision is associated with the idea of the lower the weight, the lower the strength [36]. Downgrading implies a decrease in body fat but also in muscle mass, which can have an undesirable effect on their performance and can also lead to physiological, metabolic, and immunological alterations that put their health at risk [37].

The decision as to which category a female wrestler should participate in is sometimes made by technical managers based on the criteria of club or national team representation. A common practice in order to have the best representation of the institution in competitions is to present the wrestlers in the highest possible categories. This policy entails, in some cases, distributing wrestlers of the same category into other categories, which entails an alteration of their ideal weight. The wrestler, during the competition season, is constantly forced to be at a specific weight in order to compete. Having to maintain a certain weight for a large part of the season can lead to the entrenchment of unhealthy behaviors to force this condition throughout the training cycle [35].

All these behaviors seem to be assumed by the fear of not being called in the case of not meeting the results they are expected to achieve, which may lead to a higher risk for the development of an ED [9]. The female wrestlers in our study constantly gain and lose weight to adjust it to the competition category. The number on the scale influences many female wrestlers by relating it directly to their self-image [38]. There seems to be, therefore, clear feedback between self-perception and information coming from the environment. This topic is developed in the last section.

### 4.2. Weight Control Procedures

In the discourse of practically all the interviewees, concern emerged about not knowing how to lose weight in a healthy way. Some of them tried to lose weight alone, and others left it in the hands of people who did not have the necessary knowledge to carry out weight loss. This fact was reinforced by the lack of control in the practices that was usually used [6].

One of the most interesting ideas arising from these results is that inexperienced female wrestlers often copy and repeat the strategies and behaviors of experienced wrestlers who, in turn, copied these behaviors from other wrestlers or from their own coaches [39]. It appears that these bad habits have been going on for a long time in the world of wrestling and other combat sports [40].

The most used methods seem to be the extreme restriction of food at certain times prior to weighing, vomiting to avoid weighing more, the use of laxatives and diuretics, the adjustment of menstruation, etc. [12]. Evidently, after rapid weight loss, all the female wrestlers participating in this study explained that they suffered a rebound effect that aggravated the problem [41]. This led us to understand that recurrent cycles between rapid weight loss followed by binge eating and resulting weight gain appeared to be an important factor in the development of eating disorders in weight class sports [42].

The acquisition and maintenance of unhealthy or inappropriate habits is linked to the context, which we could call the “wrestling culture”. Female athletes justify these behaviors because the context in which they find themselves normalizes these behavioral patterns, downplaying their importance and seriousness and regularizing them, and they even function as predisposing factors. It seems that, within this “wrestling culture”, the internalization and concealment of unhealthy behaviors is encouraged and generalized. This idea is supported and also appeared in the qualitative study that Papathomas and Lavallee (2010) [8] conducted with athletes who had suffered an ED; it was revealed that most athletes felt ashamed of their illness and, fearing further stigmatization, took various measures to hide it. Similarly, Peterson et al. (2008) [43] found that, in patients with bulimia, the need to preserve dignity and avoid stigmatization was so great that a “double life” was led in which dishonest concealment strategies were employed.

Other research has found that feelings of shame and the fear of stigmatization are significant barriers to seeking help [44] and to fully disclosing the situation in treatment settings [45]. This stigmatization may be accentuated in a sports environment where mental toughness is valued both culturally and in terms of individual identity [8].

### 4.3. Role of Reference Models

The idea of the pursuit of an athletic or performance ideal may be related to eating disorders, but it appears that coaches, parents, and teammates may contribute to reinforcing this relationship. The importance of relationships with parents, coaches, and teammates is highlighted in the literature [46]. To probe the topic further, in the results section, these individuals were divided into two environments, sports and social.

In the sports environment, it was mainly highlighted that the participants trusted the coaches, but the coaches were not able to identify disordered eating behaviors or even encouraged them [47]. Pulkkinen (2001) [48] found that the main sources of information on weight loss promotion for professional female judo athletes were their coach (40%) and fellow or former female judo athletes (35%). According to Díaz (2020) [47], athletes trust coaches to modify weight, thus supporting the results obtained in this study. In addition, we added another idea that emerged regarding coaches, and it is that some are not able to identify disordered eating behaviors or even go as far as to encourage them.

As explained in the previous point, inappropriate procedures are initiated when novice wrestlers follow the example of older wrestlers who, in turn, used practices learned from previous generations. There seems to be an unspoken line of transmission in certain not-too-healthy habits that is supported by the tradition of practice.

Another recurring theme in the literature on EDs is the external and internal pressures suffered by high-performance athletes. Among them, the figure of the coach was the most frequently mentioned during the interviews. The pressure that can be exerted by the coach in the world of sports can be one of the triggers of this type of disorder [9] and seems to be very significant in terms of the weight control of athletes [47]. In fact, several studies pointed out that the fact that coaches do not possess the skills for the prevention and promotion of healthy habits among their athletes can be considered a risk factor [47,49,50]. Nowicka (2013) [51] and Diaz (2020) [47] stated that some coaches do not have the necessary training and skills to detect and treat EDs in their athletes. It would be necessary to train coaches on this topic to avoid these situations in the future. One practical way to achieve this would be to add nutritional competencies in the technical curricular training of coaches [52].

Previous qualitative work also highlighted the relationship between sports interactions and social pressures, e.g., teammates modeling disordered eating behaviors at group meals or comments from parents and coaches encouraging the loss of body fat to improve sports performance with the development of disordered eating [8,53,54]. Specifically, in wrestling, group meals often take place right after competition weigh-ins, e.g., on the night of competition, wrestlers gather for dinner and overeat.

Offensive comments from peers or coaches about the physique and weight of female wrestlers appear to reinforce weight maintenance behaviors [55] and may reinforce the use of inappropriate procedures. The environment of a high-performance female athlete is often quite limited and closed. Atkinson (2011) [56] spoke of sports cultures saturated with groupthink nutritional ideologies and considered athletes with these disorders to be socially isolated within subculturally ordered norms of functioning.

These observations supported findings in the scientific literature regarding the increased risk of high-performance athletes relative to the rest of the population for body dissatisfaction traits and EDs due to peer pressure, performance, appearance expectations, and weight control protocols [57,58].

The closed environment surrounding the athlete favors that, once the ED has developed, the feeling of shame and the need to avoid judgments and criticism from people in their sporting culture will lead them to keep their behaviors secret [8]. These behaviors seem to extend to the other areas in which the affected person moves. Some of our female wrestlers explained in their interviews that they sometimes did not feel good with people outside the wrestling circle because they did not understand their situation; for example, when they could not go out to dinner with their friends because they had a competition and had to make weight, they preferred not to go out. For the athlete, a tension is generated, and is sometimes irresolvable, between the demands of high-level training and those of a cultural environment that socializes through food and drink. Despite the regimen on which an athlete’s life is based, she is expected to engage in socially normalized but unhealthy activities [59,60]. The inability to achieve a balance between the two environments leads, in many cases, to the abandonment of one of them and can lead to progressive social isolation as a consequence of the lifestyle of elite athletes [55].

Another theme that emerged strongly in the results was that all the participants expected to receive help from trained personnel (sports psychologists and sports nutritionists). We found the same idea in the study performed by Diaz (2019) [47]. To adequately address the problem, it would be very useful to have specialists in the field of health who know the characteristics of this competitive environment since athletes with eating problems often feel misunderstood by mental health professionals [61]. A controlled eating plan could help minimize the risk of using unhealthy strategies to reduce body weight [62].

Lastly, one of the strengths of this study was the use of semistructured interviews, which allowed participants to fully communicate their experiences, offering specific and unique details. Moreover, our study sample was composed of almost the entire population of female national team wrestlers from recent years. To our knowledge, this is the first qualitative study that has included high-level competitive female Olympic wrestling athletes.

However, this study has some limitations that should be mentioned and that suggest reading our results with caution. Although our participants were high-level wrestlers, it is worth mentioning that we only interviewed female athletes, so we do not know the opinion of male wrestlers or their coaches. In addition, the athletes were only Spanish, so we do not know if these problems could be generalized to the international level. Finally, it is difficult to know if the participants were completely honest or if, on the contrary, they did not tell all the details of their disorders in the interviews. Accordingly, future research could use a mixed methodology, complementing the interviews with the administration of questionnaires to assess the sense of the relationship between this sport and its corresponding weight drops and the presence of an ED. It would also be interesting to extend the study to the other sporting people, including coaches, technicians, managers or health personnel, and the other female wrestlers of other countries, to see if it is a specific problem or if it is an international problem.

## 5. Conclusions

There seem to be numerous reasons that influence female wrestlers to suffer from some type of ED, and many of them are directly related to the above-mentioned information. Consequently, EDs could be a common problem among female wrestlers. In this line, only 6 of the 30 wrestlers were discarded for not meeting the criteria of having a current or past eating disorder, meaning that almost 80% of the initial sample had some sort of food-related problem at some point in their careers.

On the one hand, the traditional culture, which we could call the “wrestling culture”, is related to unfounded practices that develop over time due to a lack of knowledge, training, and control. It seems that some athletes of this sport repeat inadequate behaviors and procedures despite the existing scientific evidence without considering how harmful they can be. This fact could be related to the wrestling culture, which encourages moving wrestlers down to the lowest weight possible because there is the belief of the lower the category, the more chances of winning because the opponents will be less strong. Unfortunately, many of these practices of losing weight are obviously inappropriate although they are normalized. Thus, most of the participants in this study showed a risk for or symptoms of having an ED, but only one of them made her ED public. The rest continue to compete without a proper clinical diagnosis or support program.

In consequence, it could be essential for federations to check how the state of this matter is with their athletes and, depending on their own results, update their training programs to the current needs of wrestlers from a safe and healthy perspective. Additionally, the incorporation of specialized personnel with specific knowledge about weight category sports, such as psychologists, nutritionists, and/or sports doctors, into teams and clubs and the consequent follow-up and elaboration of personalized diets would undoubtedly constitute an important achievement in improving the health of sportswomen and in increasing their performance.

On the other hand, old beliefs, which advocate that the methods used are the best methods because they have been used for a long time, should be discarded to open doors to new concepts of healthier weight loss methods. Moreover, the so-called “invisible training”, i.e., nutrition, rest, social relationships, etc., seems to have a direct effect on training and performance in competitions. Therefore, the wrestling staff should take it into account to improve athletes’ performance and to reduce the risk of EDs.

Lastly, the present work could be an interesting contribution to the research community, as it shows the reality faced by female wrestlers in relation to weight loss and EDs from their own perspective, facilitating the future approach of solutions and strategies aimed at improving this situation.

## Figures and Tables

**Table 1 ijerph-20-02441-t001:** Participant characteristics.

Participant	Age	Height	Age Competition Category	Weight Competition Category (kg)	Current Weight (kg)	Years of Wrestling Practice	Years of Competition	Frequency of Training (Hours per Week)	Educational Level (Finished)
L01	21	159	UNDER 23	50	53.5	18	10	14	High school
L02	26	162	SENIOR	57	63	8	7	18	University master’s degree
L03	19	174	JUNIOR	65	66	4	4	20	High school
L04	20	165	UNDER 23	57	59.5	8	7	22	Middle school
L05	17	164	JUNIOR	57	59	5	5	16	Middle school
L06	20	176	JUNIOR	76	84	8	8	18	High school
L07	24	170	SENIOR	62	64.5	15	15	20	High school
L08	22	174	UNDER 23	72	73	10	10	14	High school
L09	21	170	UNDER 23	55	56	18	10	14	High school
L10	19	166	JUNIOR	68	65.5	11	7	10	High school
L11	23	167	UNDER 23	72	76.5	13	11	6	High school
L12	18	162	JUNIOR	59	65.5	2	2	24	Middle school
L13	23	169	UNDER 23	53	57.5	12	10	12	Middle school
L14	21	165	UNDER 23	65	69	12	10	14	High school
L15	26	158	SENIOR	65	65	14	12	20	University Degree
L16	21	164	UNDER 23	68	73	12	9	20	High school
L17	17	165	JUNIOR	53	54	5	4	12	Primary school
L18	16	159	JUNIOR	50	51	13	4	20	Middle school
L19	18	176	JUNIOR	76	83.5	6	6	6	Middle school
L20	20	154	UNDER 23	50	50.9	8	8	18	High school
L21	24	158	SENIOR	50	55.5	8	7	20	University degree
L22	22	157	UNDER 23	59	62	9	9	25	High school

**Table 2 ijerph-20-02441-t002:** Emerging themes and subthemes of the analysis.

1. Reasons for Weight Reduction	2. Inadequate Procedures	3. Role of Reference People
-Limitation of participation in international championships	-Lack of adequate information	-Sports environment
-Body changes: development and body image	-Risky practices	-Social and family environment

## Data Availability

Not applicable.

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
