# Peer review of "Training Conditions and Psychological Health: Eating Behavior Disorders in Spanish High-Performance Women’s Olympic Wrestling Athletes—A Qualitative Study"

_ijerph, 2023, doi:10.3390/ijerph20032441_

Round 1
Reviewer 1 Report
Thank you for the opportunity to review this paper which examined the factors that influence the occurrence of behaviors related to possible eating disorders in female elite wrestlers.
I found it interesting to read, but I have some concerns about the methodological approach. I am not sure that a phenomenological approach is adequate for the research question. Given the phenomenological approach, I would have liked to get more insight into the experiences and personal meanings of the participants. Another concern relates to extrapolation and generalized statements that are not valid because the sample studied is not representative and the study is qualitative in nature. The presentation of the results appeared superficial. They were hardly put into the context of the current scientific literature. Major revisions are needed here. I suggest you really focus on your results and discuss their significance in light of the current evidence.
Of great concern are the outdated and often erroneous references. Consequently, the current work could not be discussed in light of the current evidence, which significantly diminishes the contextualization of this work in the existing body of evidence. It is essential that the references are uptdated and corrected.
Specific suggestions:
Introduction
(31) The current ICD-11 and DSM 5 classification systems identify more than two specific eating disorders. Both ICD-11 and DSM 5 list binge eating disorder as an official eating disorder. Please update accordingly.
(70) The sentence: Regarding gender… is missing a citation.
(71) The sentence: It seems that the incidence… is missing a citation. Did you mean incidence or prevalence?
(73-76) Please taper down the use of definitive language.
Materials and methods
Study Design
Please explain the methodological approach. What is meant by the phenomenological paradigm/approach? IPA? Explain why it is suited for the research question.
According to my understanding the main objective of this study was to identify the factors that influence the occurrence of behaviors related to possible eating disorders. Given the main objective, it is not clear to me why a phenomenological approach was chosen, as this primarily examines the meaning and significance of experience. Grounded theory would seem more appropriate to address the primary objective.
Participants:
(99) How were participants selected? how were they approached? Why did you end up with 22 participants? Were participants recruited until no new relevant knowledge was obtained? (data saturation)
(102) Please provide more participant characteristics (educational background, years of wrestling, frequency of trainings…), so readers can consider the relevance to their own situation/field. Please provide a table that includes the above information as follows:
|
Participant pseudonym |
Age |
Eating disorder (yes/no) |
Symptoms and background information |
|
L01 |
.. |
.. |
.. |
Material and data collection:
There are almost no characteristics reported about the interviewer and researchers. In qualitative research, researchers are closely involved with the research process and participants and therefore cannot completely avoid personal bias. It is therefore important to clarify their identity, gender, occupation, experience… This makes it easier to understand how these factors might interact with the interpretation.
(109) Which authors developed the interviews? What were the researcher’s credentials? MD, Psychologist, Sports scientist? Were the researchers male or female?
(121) Which author conducted the interview? What were the author’s credentials? MD, Psychologist, Sports scientist? Was the interviewer male or female? What experience or training did the interviewer have? Was a relationship established prior to study commencement?
(126) Did you use audio or visual recording?
(127) What exactly was explained to them? What purpose was given?
(142) Were field notes made during or after the interview?
(147) Were transcripts returned to the participants?
(147) Who made the transcripts? A single author or multiple authors?
(151) How many data coders coded the data?
(158) Did participants give feedback on the research findings?
Results:
It is noticeable that not all participants were quoted to illustrate the results. In fact, 9 out of 22 participants were not quoted at all, while other participants were quoted several times (especially L05 and L06). It is also not clear which participants contributed to which theme. It would be helpful to add the column "Participants contributing" to Table 1 (Emerging themes and subthemes of the analysis). For example as follows:
|
Themes |
Subthemes |
Participants contributing |
|
Reasons for weight reduction |
Limitation of participation in international championships Body changes: development and body image |
L01, L07, L11, L19, L22 |
(182) L011? Not L11?
(221) italic font is missing
Discussion:
Throughout the discussion, the use of definitive language needs to be toned down. This is not a representative sample, so generalizability of statements is not valid. It doesn't seem valid to me to keep referring to “wrestlers” when referring to the participants in the study. That suggests a generalizability that is not applicable here. Please focus more on your findings and try to discuss them in the context of the current literature. The discussion should be balanced and based on current literature. Major revisions are needed here. In particular, a discussion with diverging results from the literature is also necessary. Unfortunately, many references used are either no longer up to date or do not support the statements.
Reason for weight reduction:
The drive for thinness has been identified as a major risk factor for the development of eating disorders in elite athletes. Your findings also seem to point in that direction. Discussion of your findings with the current literature on drive for thinness would contribute to the paper.
(362) The Picard study cited here does not support this statement. The picard study indicated that athletes at higher levels of competition showed more signs of pathological eating. The study investigated the influence of levels of competition on disordered eating, not the practice of dropping in weight class.
(363) The Milligan and Pritchard study cited here does not support this statement. In fact, this study found that women in non-lean sports displayed the highest level of disordered eating behavior and body dissatisfaction. So the exact opposite of what is stated here.
(385 – 387) …seems to be associated with certain psychological traits…seems to be traits associated with certain psychological conditions.
These statements are extremely non-specific, so it is not clear what is meant. Please elaborate
(382) The reference given here is a guide for parents and hardly an adequate citation for the statement. There are certainly better references for the well-known association between perfectionism and disordered eating/eating disorders.
(394) Reference 32 does not support the statement. It’s a review paper from 1999 that doesn’t clearly discuss the role of body dissatisfaction in the development of eating disorders in athletes. I'm certain that a more current reference would be more appropriate for this. Especially since the role of body dissatisfaction in the development of eating disorders in elite athletes does not seem quite as clear as in the general population.
(410 – 412) Please temper down the use of definitive language. The statement here is not supported by an original paper but by a book. This is unfortunate, since it is not transparent on which data/studies the statement is based. The development of eating disorders is multifactorial, to single out one particular factor as central does not seem justified.
(415) The reference cited here does not support your statement. You cite the DSM-4, which makes no specific statement about wrestlers. In addition, the DSM 5 has been available for several years, and diagnostic criteria should be cited with the DSM 5. Please provide a reference that shows that the number on the scale is directly related to self-image in wrestlers.
Inadequate Procedures
(435) The reference cited here does not support your statement. The study cited here found that a 3 hour rehydration did not improve glycogen levels or strength performance after a four-day period of food and liquid intake reductions to lose weight.
(436) The recurring cycles between rapid weight loss followed by binge eating and resulting weight gain appears to be an important factor in the development of eating disorders in weight class sports. Please discuss this with the current literature.
(436) There is no reference to negation in the results. Nevertheless, it takes a prominent stand here. If we look at the statements of L04 and L06 (I have a problem and I need help..., I knew it was wrong, but what was wrong with me?..), then it seems, on the contrary, that the participants were well aware of their problems.
(440-449) This section would benefit from you staying closer to your data. The focus is on inadequate procedures and their importance in an athletic context, not on shame and the feeling of not being sick enough. That's not mentioned in the results.
(453-458) This could be solved… Copying strategies and behaviors from experienced wrestlers is an interesting finding. This should be discussed in the social system of high performance sports. Here, solutions are proposed too quickly. This is not the focus of a phenomenological approach.
The role of reference people
(462) There references (17 and 42) cited here do not support your statement. The study by Sundgot-Borgen investigated the prevalence of eating disorders in elite athletes, not the influence of the athletic environment on the development of eating disorders. The study by Woods did not include elite athletes.
(470 – 515) Again, this section would benefit from you staying closer to your data. In the results sections, you presented two themes (sports environment and social and family environment) and mainly highlighted that participants trust coaches, but coaches are not able to identify disordered eating behaviors or even encourage them. It was also emphasized that all participants hoped for help from trained personnel (sports psychologists, sports nutritionists). Regarding the family environment, you highlighted that the family wanted to help the participants, but this help was not always seen as beneficial because the family was not aware of the sport-specific requirements. These findings should now be compared and discussed with the current literature. You mention studies that showed that family members of elite athletes motivated them to lose fat (484). However, you have presented results that show that the family wanted to help the participants. These differences between your results and the studies presented need to be discussed.
(474) The reference cited here does not seem to support your statement. The study you cited examined the relationship between eating disorders and perfectionism and not the influence of coaches on weight control behaviors.
(476) The reference cited here does not support your statement. The study by Ferrand examined unhealthy compensatory behaviours in synchronized swimmers. It did not examine the level of knowledge of coaches.
Conclusions
(517) Several studies have shown that eating disorders are very common among elite athletes in leanness sports. Eating disorders are therefore not more common than we realize.
(517-518) The fact that most female wrestlers showed some sort of food related problem at some point in their careers is clearly due to the inclusion criteria of the study. After all, only participants who had a current or past eating disorder were included. It is therefore logical that all participants would have food related problems. Again, your findings are not generalizable.
(521-564) The conclusions go far beyond what the data actually allow. Several major strategies for change are suggested here, but this was not the subject of the study at all and must not be inferred, since this is a qualitative study on a non-representative sample. Instead, the conclusion should also be limited to the data obtained during the study. Major revisions are needed here.
Reviewer 2 Report
The paper objective is to determine the factors that affect the occurrence of behaviors related to possible eating disorders in Spanish high-performance Olympic wrestling athletes. It is considered a necessary work from the qualitative approach.
1) It is necessary to establish the research limits
2) It is recommended to optimize the "Conclusions" section, it is very extensive, and statements many can be detailed in the Results and Discussion sections.
3) The statement in the "Conclusions" section, available between lines 550-554 and 555-561, are typical of the "Results" and "Discussion" sections. The affirmations must be placed in the mentioned sections.
